# Influence of Zinc Content on the Mechanical Behaviors of Cu-Zn Alloys by Molecular Dynamics

**DOI:** 10.3390/ma13092062

**Published:** 2020-04-29

**Authors:** Heung Woon Jang, Jung-Wuk Hong

**Affiliations:** Department of Civil and Environmental Engineering, Korea Advanced Institute of Science and Technology, Daejeon 34141, Korea; hwjang90@kaist.ac.kr

**Keywords:** molecular dynamics simulation, copper alloy, zinc alloy additive, large-scale atomic/molecular massively parallel simulator, modified embedded atom method

## Abstract

The mechanical properties of copper alloys containing various ratios of zinc are evaluated using molecular dynamics (MD) simulations to determine the impact of the different zinc concentrations. The modified embedded atom method (MEAM) parameters for copper were established in the 1990s; however, the MEAM potential parameters for zinc, with an axial ratio >1, were recently proposed. In this research, the MD models of the copper alloys with various zinc contents are constructed using the MEAM potential parameters for zinc. Tensile test simulations are also conducted. The strain rate effects of the alloys are evaluated at four different strain rates, and the variations in the tensile strengths and Young’s modulus are investigated. The proposed procedures have significant potential applicability for simulating a variety of zinc-containing alloys.

## 1. Introduction

Copper alloys have been widely utilized in various applications, such as in wires, water pipes, and roof systems, due to their high thermal and electrical conductivities. Pure copper is mechanically soft and unsuitable for numerous applications. However, copper can be hardened when used as an alloy, which greatly enhances its applicability. Consequently, various copper alloys have been developed and classified according to the principal alloying elements (Table 1).

Brass—a copper alloy with Zn as the main additive—is utilized in various applications because of its excellent rigidity and relatively low melting point. However, there is limited research regarding the mechanical behaviors of brass at the molecular level. Therefore, this study is conducted to address the research gap by investigating the aforementioned behaviors relative to the zinc content for various strain rates. The effect of the zinc content on the mechanical behaviors of the copper alloy is also observed.

In order to perform the molecular dynamics (MD) analysis, it is essential to understand the potential function. Originally, the pair potential was used for the MD simulations; however, there were complications when implementing the metallic bonds [1]. To solve this problem, the embedded atom method (EAM) based on the density functional theory was developed by Daw and Baskes, and a hydrogen embrittlement phenomenon, which was difficult to simulate with the pair potentials, was also utilized [2]. Baskes et al. developed a modified embedded atom method (MEAM), which allows the angular dependency to be included in the electron density of the EAM [3].

Several MD studies for copper were previously undertaken. Ye et al. conducted the MD simulations of the nanometric cutting of a single-crystal copper [4]. Zhang et al. studied the melting behaviors of copper nanorods by employing MD simulations with the EAM potential [5]. Zhang and Jiang investigated atomistic mechanisms for the temperature-induced crystallization of amorphous copper by using MD simulations [6]. The MEAM potential for zinc was recently developed to perform the molecular analysis of the mechanical properties of an aluminum 7075 alloy [7]. However, due to the recent development, the MD simulation of the copper alloy, which is the principal alloying element of zinc, has not advanced. We performed an MD simulation on the effect of strain rate on the mechanical behaviors of the copper-zinc alloys containing different concentrations of zinc and observed the variations.

## 2. Theoretical Basis

We use the MEAM potential parameters for zinc, based on the EAM potential parameters for zinc [8] and the MEAM potential parameters for copper [3,9,10]. In addition, the dislocations of the metals with face-centered cubic (FCC) structures are investigated to determine the failure properties of the alloys.

### 2.1. Embedded Atom Method (EAM)

The proposed MEAM potential model for zinc is based on the EAM potential function of the zinc developed by Romer [8]. The energy term *E* is written as
(1)$$E=\sum_{i=1}^{N}F_{i}\left[\rho_{h,i}\left(r_{i}\right)\right]+\frac{1}{2}\sum_{i,j(i\neq j}^{N}\phi_{ij}\left(r_{ij}\right),$$
where $F_{i}$ are the empirically obtained embedding functions, $\phi_{ij}\left(r_{ij}\right)$ are the repulsive pair potentials, and $\rho_{h,i}$ are the local electron densities at the positions $r_{i}$. The local electron densities $\rho_{h,i}$ calculated by summing up the electron density distributions $\rho_{j}^{at}$ of all the atoms within the interaction range are expressed by
(2)$$\rho_{h,i}\left(r_{i}\right)=\sum_{j\neq i}^{N}\rho_{j}^{at}\left(r_{ij}\right).$$

The wave function of the $4_{s}$-electrons $\psi_{4s}$ is used to obtain the electron density distribution of zinc as
(3)$$\rho_{j}^{at}\left(r\right)=2\cdot {\left|\psi_{4s}\right|}^{2},$$
where the detailed s-character is negligible because Zn $\left(3d^{10}4s^{2}\right)$ is a closed-shell system [8]. To reduce the complication of the model, a two-parameter exponential function is utilized for the wave function [11] as
(4)$$\rho_{j}^{at}\left(r\right)\equiv f_{e}e^{-\beta \left(\frac{r}{r_{e}}-1\right)},$$
where $f_{e}$ and β are the parameters implemented by Romer, and $r_{e}$ is the equilibrium interatomic separation distance. In the classical EAM, the repulsive pair potentials $\phi_{ij}$ are calculated from the Coulomb interaction as
(5)$$\phi_{ij}\left(r_{ij}\right)=\frac{Z_{i}\left(r_{i}\right)\cdot Z_{j}\left(r_{j}\right)}{r_{ij}},$$
where $Z_{i}$ are the effective core charges and are written as
(6)$$Z_{i}\left(r_{i}\right)=Z_{0}-\int_{0}^{r}\rho^{at}\left(r\right)\cdot 4\pi {\left(r-r_{i}\right)}^{2}dr,$$
and where the plain core charge $Z_{0}=2$. By using a two-parameter expotential function Equation (Equation 3) is simplified to Equation (Equation 4). Similarly, the effective core charge $Z_{i}\left(r_{i}\right)$ in Equation (Equation 6) is simplified as
(7)$$Z_{i}\left(r_{i}\right)\equiv Z_{e}\cdot e^{-\alpha \left(\frac{r-r_{i}}{r_{e}}-1\right)},$$
where α and $Z_{e}$ are the parameters provided by Romer.

### 2.2. Modified Embedded Atom Method (MEAM)

Similar to the EAM, it is also essential to calculate the total energy in Equation (Equation 1) for the MEAM. Local electron densities $\rho_{h,i}$ are also calculated by summing up the spherically averaged atomic electron densities, as in Equation (Equation 2). However, the MEAM modifies the $\rho_{h,i}$ to account for the angular dependency as
(8)$$\rho_{h,i}=\frac{\rho_{h,i}^{\left(0\right)}}{\rho_{i}^{0}}G_{i}\left(\Gamma_{i}\right),\Gamma_{i}=\sum_{k=1}^{3}t_{i}^{\left(k\right)}{\left(\frac{\rho_{h,i}^{\left(k\right)}}{\rho_{h,i}^{\left(0\right)}}\right)}^{2},$$
where $G_{i}\left(\Gamma \right)$ are the functions determined by the element type and $\rho_{i}^{0}$ are the composition-dependent electron densities. The partial electron densities $\rho_{h,i}^{\left(k\right)}$ are written as
(9)$$\begin{array}{c} \rho_{h,i}^{\left(0\right)}=\sum_{j\neq i}^{}\rho_{j}^{a\left(0\right)}\left(r_{ij}\right)S_{ij},\\ {\left(\rho_{h,i}^{\left(1\right)}\right)}^{2}=\sum_{\alpha =1}^{3}{\left[\sum_{j\neq i}^{}\rho_{j}^{a\left(1\right)}\left(\frac{r_{ij\alpha }}{r_{ij}}\right)S_{ij}\right]}^{2},\\ {\left(\rho_{h,i}^{\left(2\right)}\right)}^{2}=\sum_{\alpha =1}^{3}\sum_{\beta =1}^{3}{\left[\sum_{j\neq i}^{}\rho_{j}^{a\left(2\right)}\left(\frac{r_{ij\alpha }r_{ij\beta }}{r_{ij}^{2}}\right)S_{ij}\right]}^{2}-\frac{1}{3}{\left[\sum_{j\neq i}^{}\rho_{j}^{a\left(2\right)}\left(r_{ij}\right)S_{ij}\right]}^{2},\\ {\left(\rho_{h,i}^{\left(3\right)}\right)}^{2}=\sum_{\alpha =1}^{3}\sum_{\beta =1}^{3}\sum_{\gamma =1}^{3}{\left[\sum_{j\neq i}^{}\rho_{j}^{a\left(3\right)}\left(\frac{r_{ij\alpha }r_{ij\beta }r_{ij\gamma }}{r_{ij}^{2}}\right)S_{ij}\right]}^{2}-\frac{3}{5}\sum_{\alpha =1}^{3}{\left[\sum_{j\neq i}^{}\rho_{j}^{a\left(3\right)}\left(\frac{r_{ij\alpha }}{r_{ij}}\right)S_{ij}\right]}^{2}, \end{array}$$
where $S_{ij}$ are the shape factors for various structures and $\rho_{i}^{a\left(k\right)}\left(r_{ij}\right)$ are expressed as
(10)$$\rho_{i}^{a\left(k\right)}\left(r_{ij}\right)=\rho_{i0}\exp\left[-\beta_{i}^{\left(k\right)}\left(\frac{r_{ij}}{r_{i}^{0}}-1\right)\right].$$

It should be noted that, if *k* is 0, the equations of the electron density distribution of the EAM in Equation (Equation 2) and of the MEAM in Equation (Equation 8) are same.

### 2.3. Dislocations in FCC Lattices

In a previous study, we developed a method for the quantitative measurement of the number of dislocations in the FCC lattices of the aluminum alloy [7]. Since copper also possessed an FCC lattice structure, we quantitatively measured the dislocation that occurred in the copper alloy by using the proposed method. When dislocation occurred in the FCC lattice, it followes the process depicted in Figure 1.

Figure 1a illustrates the structure of the close-packed FCC lattice with an ABCABC arrangement. Figure 1b,c show the cases of perfect dislocation and partial dislocation in FCC, respectively. When partial dislocation occurred, the FCC lattice is transformed into the hexagonal closed packed (HCP) lattice, as shown in Figure 1d. In this study, the magnitude of dislocation is estimated by measuring the ratio of HCP lattices transformed from the FCC lattices.

## 3. Molecular Dynamics Simulation

In this study, the MD analyses of copper-zinc alloys with different zinc contents are performed using the Large-Scale Atomic/Molecular Massively Parallel Simulator (LAMMPS) program developed by Sandia National Laboratories [12].

### 3.1. Potential Model

A potential model that takes both copper and zinc into consideration is required to simulate the copper-zinc alloy model. The MEAM potential parameters were established for copper in previous studies [3,9,10]. However, those potential parameters for zinc were only recently developed because the axial ratio (c/a) of zinc is greater than 1 [10], where *c* and *a* are the actual lengths of the crystallographic axes, respectively. Therefore, we proposed a novel method for applying the verified EAM potential parameters of zinc into the MEAM potential parameters [7]. The MEAM parameters used for copper and zinc are listed in Table 2.

### 3.2. Atomistic Model

Two different models are developed to evaluate the effect of zinc on the mechanical behaviors of copper alloys. The first model contains 97.5% Cu and 2.5% Zn, whereas the other contains 95% Cu and 5% Zn. If more zinc is included in the model, the more computational time is required to stabilize the model. In order to make the analysis and research efficient, the zinc contents are minimized to the extent that zinc influences the mechanical behaviors of the alloy model. The original geometrical configurations of the atomistic models are illustrated in Figure 2.

The copper-zinc alloy model with 2.5% Zn contains 33,158 copper atoms and 1013 zinc atoms, whereas the alloy with 5% Zn contains 32,688 copper atoms and 2012 zinc atoms. The models of the copper-zinc alloys are prepared by using the methods to model the alloys proposed by previous studies. Several studies found that an unknown solid structure is formed under the temperature of the model increases to 800 K with zero pressure [13,14]. Therefore, the zinc atoms are randomly distributed in the copper FCC lattices as performed in the previous studies. During the distribution of the zinc atoms, if the added zinc atoms are too close to the previously positioned copper atoms, the copper atoms are replaced with zinc atoms. The atomistic models are constructed as a cube of length, 20Δx. The unit, Δx, is the equilibrium lattice parameter of copper (3.62 Å).

### 3.3. Simulation Methods

We perform simulations by applying tensile loads with different strain rates to each model. A stabilization process is conducted for 20 ps ($10^{-12}$ s) before the tensile loads are applied. The stabilization process is performed for a total of 20,000 steps with a step time of 2 fs ($10^{-15}$ s). The Nose–Hoover thermostat is used to maintain the temperature of the system at 300 K. The pressure of the system is maintained at zero bar using the Berendsen Barostat algorithm [15]. The crystallization process is performed by first increasing the temperature of the system from 300 K to 800 K and then decreasing to 300 K. The stabilization and crystallization processes are used to ensure the stability of each model. In general, mechanical behaviors in MD simulations, such as Young’s modulus and tensile strength, vary with the strain rate. Therefore, the tensile loads are applied at four different strain rates in this study. The periodic boundary conditions are implemented in three orthogonal directions (x=[100], y=[010], and z=[001]) to efficiently simulate a large domain. The periodic boundary promotes the interaction of particles across the boundary, thereby enabling them to exit from one end and to re-enter at the other [12].

## 4. Numerical Results

### 4.1. Stress–Strain Curve

The uniaxial tensile loads are applied to the models at four different strain rates. The relationships between stress and strain are obtained. The highest stress values in Figure 3 are the ultimate tensile strengths. The slope at the beginning of the graph is Young’s modulus. The stress–strain curves of the two models are compared and the tensile strength of the 2.5%-Zn model is greater than that of the 5%-Zn model.

The ultimate tensile strength increases as the strain rate increases in both models. Young’s modulus of the 2.5%-Zn model increases as the strain rate increases. However, Young’s modulus of the 5%-Zn model decreases as the strain rate increases. Young’s modulus and tensile strength of the models in accordance with strain rates are shown in Figure 4.

The results in Figure 3 and Figure 4 are in similar scales with the results of Chang and Fang’s study [16] that performed molecular dynamics simulations on copper. Considering the data in Figure 4 and using the regression technique for numerical fitting, Young’s modulus and the tensile strength of the 2.5%-Zn model are formulated as
(11)$$E=-\frac{2.08{\dot{\epsilon }}^{2}}{10^{17}}+\frac{3.14\dot{\epsilon }}{10^{8}}+46.73$$
(12)$$\sigma_{t}=-\frac{1.09{\dot{\epsilon }}^{2}}{10^{18}}+\frac{2.32\dot{\epsilon }}{10^{9}}+4.90$$
and the properties of the 5.0%-Zn model are formulated as
(13)$$E=\frac{3.37{\dot{\epsilon }}^{2}}{10^{17}}-\frac{6.84\dot{\epsilon }}{10^{8}}+49.41$$
(14)$$\sigma_{t}=-\frac{9.76{\dot{\epsilon }}^{2}}{10^{19}}+\frac{1.86\dot{\epsilon }}{10^{9}}+2.00$$

### 4.2. Transformation of the Lattices

As described in Section 2.3, the FCC lattice structures are transformed to the HCP lattice structures when partial dislocations occurs in the FCC lattice structures. Wu et al. observed that when tensile loading is applied to an FCC structure, the lattice is transformed and the twin boundary due to partial dislocation is generated, as shown in Figure 5 [7,17].

The locations of the copper atoms are extracted separately using our code and then bond-angle analysis is conducted by using the Open Visualization Tool (OVITO). Through bond-angle analysis, numbers and locations of atoms corresponding to various lattices are obtained separately. To check the distribution of HCP atoms according to the strains, cross sections of the model are plotted at every 0.16 strain interval.

The ratios of the HCP lattices increase as the strain increases in both models, as shown in Figure 6. The numbers of FCC atoms and HCP atoms are compared with the total number of atoms. We investigate this phenomenon more precisely by calculating the ratio of the FCC lattices and HCP lattices under tensile deformation.

The numbers of copper atoms that form the FCC, HCP, BCC (body-centered cubic), and other lattice arrangements are counted, and the ratios of the copper atoms that correspond to the FCC and HCP structures are calculated. As the strain increases, there is a decrease in the proportion of the FCC lattices as well as an increase in that of the HCP lattices for both models, as shown in Figure 7. In addition, the ratios of the lattices vary rapidly at the strain values between 0.1 and 0.2. A comparison of the results with the stress-strain curves in Figure 3 reveals that the range where the ratio of lattice changes rapidly and the range where failure occurs are in agreement. The provided evidence that the FCC lattices are transformed into the HCP lattices when failure occurs, and that the changes in the ratios are closely related to the mechanical properties of the alloy. It is observed that the sudden change in the tensile stress of the 2.5% alloy is due to the failure of the FCC lattice structures. Comparing the two models, the percentage of FCC lattice at the end of stabilization is significantly higher for the 2.5%-Zn alloy than for the 5%-Zn alloy. Overall, such a difference in the FCC lattice ratio results in changes in the tensile strength of alloys.

## 5. Conclusions

We analyze the mechanical behaviors of the copper–zinc alloys with different ratios of zinc by using the recently developed MEAM type potential model for zinc [7] by MD simulations. Copper–zinc alloys, also called brass, are widely used in various fields. However, simulations at the molecular level have only been performed recently because the MEAM model was not developed to analyze copper and zinc. We prepared copper–zinc alloy models that contains 97.5% Cu, 2.5% Zn and 95% Cu, 5% Zn. These prepared alloy models are used to verify our proposed MEAM model for zinc and to analyze the effect of zinc content on the mechanical behaviors of the copper-zinc alloys. The tensile test simulations are conducted on the models at four different strain rates by MD simulation. The results of the two different simulations are compared. The effects of the strain rate on the mechanical behaviors are analyzed and the tensile strength of both models increased with increasing strain rate. However, Young’s modulus of the 2.5% Zn-alloy model is not impacted by the strain rate, whereas that of the 5% Zn-alloy model decreases with increasing strain rate. These results are related to the ratio of the lattice structures; hence, the ratios of the FCC and HCP lattices are calculated. As the strain is increased in both models, the proportion of the lattices decreases and increases for FCC and HCP, respectively. Sudden changes in the ratios of the lattices occur at strain values between 0.1 and 0.2, which is consistent with the strains at which failure occurs in each model. The phenomenon proves that the FCC structure of copper is converted to the HCP structure when failure occurs. In the comparison of the two models, the proportion of the FCC lattices at the end of stabilization is significantly higher for the 2.5%-Zn alloys than the 5%-Zn alloys. It is assumed that the differences in the FCC lattice ratio results in variations in the tensile strength. In future studies, we will explore the effect of the zinc content on the mechanical behaviors of various zinc-containing alloys at the molecular level by the method we proposed in this study.

## Figures and Tables

**Figure 1 materials-13-02062-f001:**
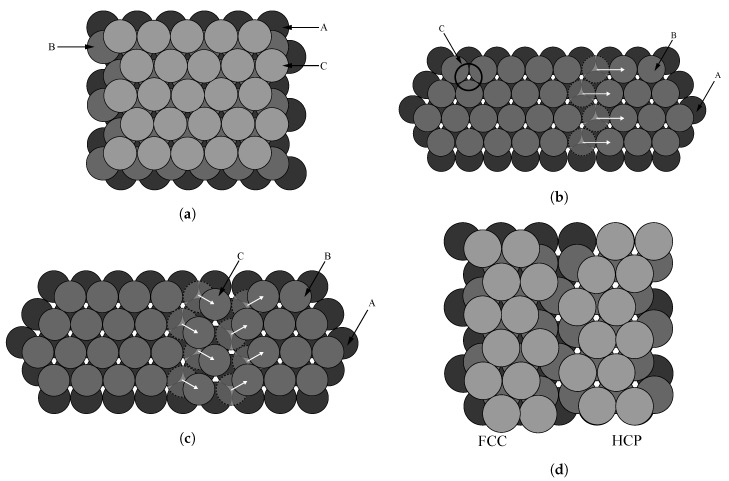
The FCC lattice with: (**a**) ABCABC stacking arrangement, (**b**) perfect dislocations, (**c**) partial dislocations, and (**d**) hexagonal close-packed (HCP) lattices due to partial dislocations [7].

**Figure 2 materials-13-02062-f002:**
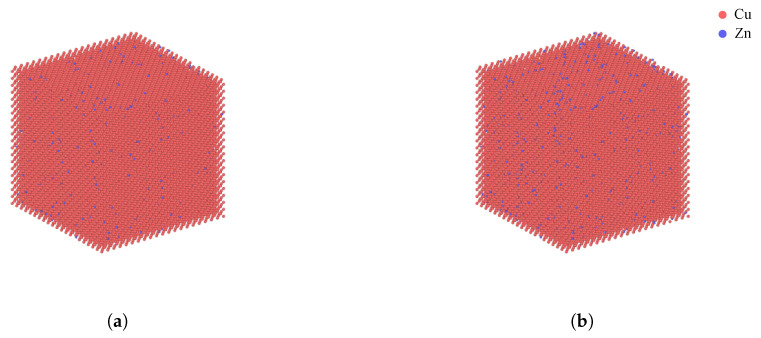
Original configurations of the Cu-Zn alloys with zinc contents of (**a**) 2.5% and (**b**) 5%.

**Figure 3 materials-13-02062-f003:**
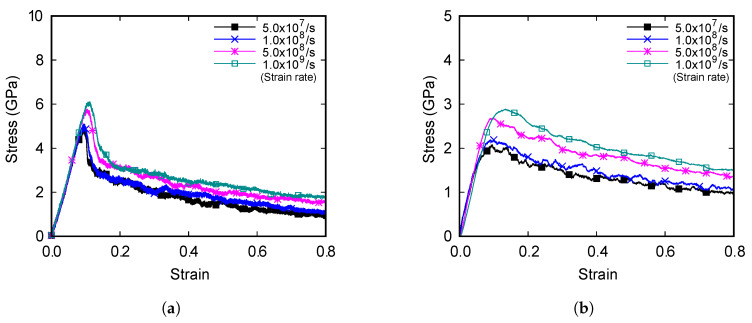
Stress-strain curves of the Cu-Zn alloys with the zinc content of (**a**) 2.5% and (**b**) 5%.

**Figure 4 materials-13-02062-f004:**
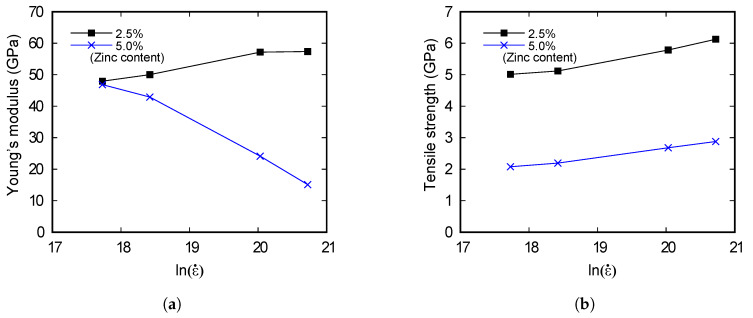
(**a**) Young’s modulus and (**b**) the tensile strength of the models in accordance with strain rates.

**Figure 5 materials-13-02062-f005:**
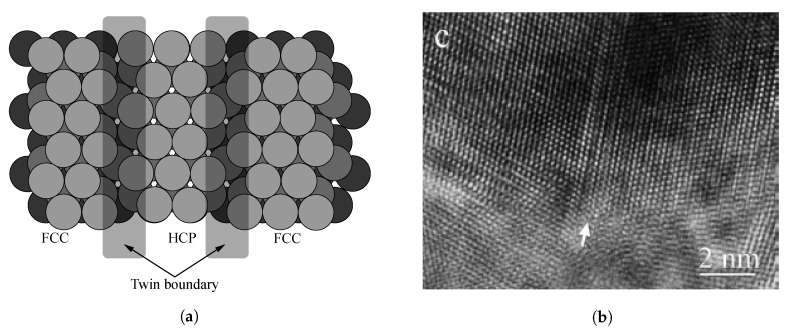
(**a**) Diagram of the twin boundary caused by partial dislocation and (**b**) twin boundary in an FCC lattice due to the tensile loading observed in reference [7,17].

**Figure 6 materials-13-02062-f006:**
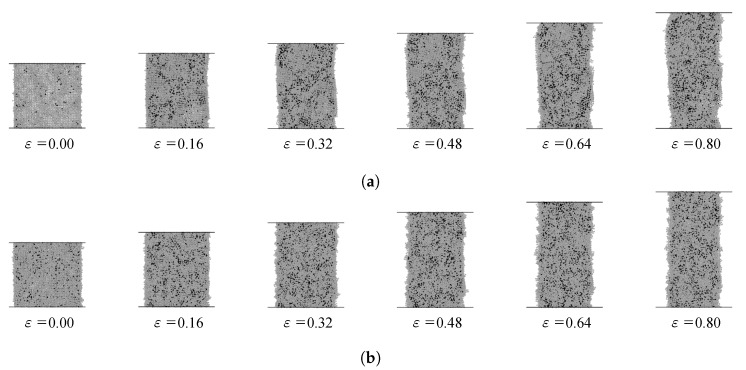
Distribution of the HCP atoms (black dots) in the Cu-Zn alloy models with the zinc content of (**a**) 2.5% and (**b**) 5%.

**Figure 7 materials-13-02062-f007:**
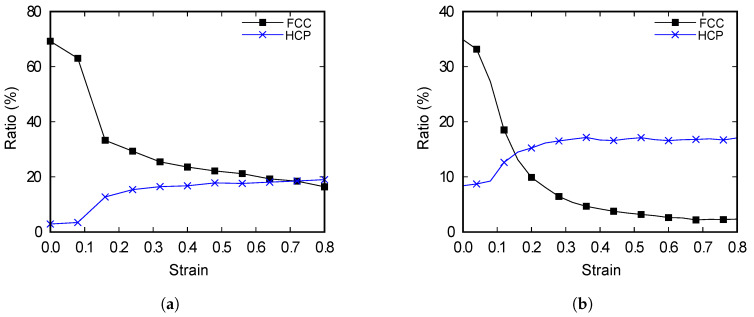
Lattice ratio–strain curve for copper alloy models with zinc contents of (**a**) 2.5% and (**b**) 5%, respectively.

**Table 1 materials-13-02062-t001:** Types of copper alloys.

Type	Principal Alloying Element
Brass	Zinc (Zn)
Phosphor bronze	Tin (Sn)
Aluminum bronzes	Aluminum (Al)
Silicon bronzes	Silicon (Si)
Cupronickel, nickel silvers	Nickel (Ni)

**Table 2 materials-13-02062-t002:** MEAM parameters for copper alloys [3,8,9,10].

Elem.	$E_{c}$ (eV)	$A_{0}$ (Å)	*A*	α	$\beta^{\left(0\right)}$	$\beta^{\left(1\right)}$	$\beta^{\left(2\right)}$	$\beta^{\left(3\right)}$	$t^{\left(0\right)}$	$t^{\left(1\right)}$	$t^{\left(2\right)}$	$t^{\left(3\right)}$
Cu	3.54	3.62	1.07	5.11	3.63	2.2	6.00	2.20	1.0	3.138	2.494	2.95
Zn	1.55	2.67	1.00	4.24	5.76	0	0	0	1.0	0	0	0

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
