# Peer review of "Influence of Zinc Content on the Mechanical Behaviors of Cu-Zn Alloys by Molecular Dynamics"

_materials, 2020, doi:10.3390/ma13092062_

Round 1
Reviewer 1 Report
The paper “Evaluation of the effects of zinc content on the mechanical behaviors of Cu-Zn alloys by molecular dynamics” evaluated the mechanical properties of copper alloys containing various ratios of zinc. The influence of strain rate also have been studied. To achieve these objectives, the authors have developed a modified embedded atom method (MEAM).
To carry out their study, the authors rely on the results of research previously published by other authors and by themselves.
In the introduction section, a review of these works is made, establishing, in addition, the specific objectives of their research. In the theoretical basis and MD simulation sections the authors relate the previously established models with their new proposal. Finally, the results section presents and explains the results obtained and the contributions of the authors. The conclusions section summarizes everything presented above
I consider it a very interesting job. The presentation and structure of the paper is very correct, so, in my opinion, it can be published in the present format.
Reviewer 2 Report
Paper describes the molecular dynamics simulation of properties of two Cu-Zn alloys. The effect of the addition of Zn was stated. Manuscript is well-written however I have a few remarks:
Please explain at first appearance, all the abbreviations and acronyms in the paper both, e.g., MD. Also, provide the full phrase in the heading of section "2.1. EAM".
In section 3.2, please explain/ justify why the 2.5 and 5% Zn addition were investigated.
In section 3.3., please explain the time-unit: 20ps and 2fs - simply add bracket comment after the unit. Simply, it is unclear at first glance.
The presented in figs 3 and 4 mechanical properties should be commented or compared with the properties of Cu-alloys (nominal) given by the literature.
4.2 section, L125: please add the methodology of related to the "Open Visualisation Tool" in "methodology section".
Reviewer 3 Report
In this paper the mechanical behaviors of the copper-zinc alloys with different ratios of zinc by using the recently developed MEAM type potential model for zinc by MD simulations is investigated. The work is done systematically and in a scientific manner and the results are interesting with respect to academic.
The subject is worthy of investigation and the conclusions are supported by the results. The modeling is properly addressed and the results of the developed model are in good agreement: a comparison of the results with the stress-strain curves reveals that the range where the ratio of lattice changes rapidly and the range where failure occurs are in agreement.
The research has an appropriate design using adequately methods. The results are clearly presented and the conclusions are supported by the results.
The simulation was performed for 4 different strain rates, which whose values are quite high, unusual for industry. What was the criterion for choosing these high values instead common values? And for the common strain rates, how could you appreciate the influence of the zinc content on mechanical behavior of Cu-Zn alloys?
